# Fractional Step Scheme to Approximate a Non-Linear Second-Order Reaction–Diffusion Problem with Inhomogeneous Dynamic Boundary Conditions

**Constantin Fetecău [1] and Costică Moroşanu [2,\*]**

1 Academy of Romanian Scientists, 54 Splaiul Independentei, 050094 Bucharest, Romania
2 Department of Mathematics, "Alexandru Ioan Cuza" University, Bd. Carol I, 11, 700506 Iaşi, Romania
\* Correspondence: costica.morosanu@uaic.ro

**Abstract:** Two main topics are addressed in the present paper, first, a rigorous qualitative study of a second-order reaction–diffusion problem with non-linear diffusion and cubic-type reactions, as well as inhomogeneous dynamic boundary conditions. Under certain assumptions about the input data: $g_d(t,x)$, $g_{fr}(t,x)$, $U_0(x)$ and $\zeta_0(x)$, we prove the well-posedness (the existence, a priori estimates, regularity and uniqueness) of a solution in the space $W_p^{1,2}(Q) \times W_p^{1,2}(\Sigma)$. Here, we extend previous results, enabling new mathematical models to be more suitable to describe the complexity of a wide class of different physical phenomena of life sciences, including moving interface problems, material sciences, digital image processing, automatic vehicle detection and tracking, the spread of an epidemic infection, semantic image segmentation including U-Net neural networks, etc. The second goal is to develop an iterative splitting scheme, corresponding to the non-linear second-order reaction–diffusion problem. Results relating to the convergence of the approximation scheme and error estimation are also established. On the basis of the proposed numerical scheme, we formulate the algorithm **alg-frac_sec-ord_dbc**, which represents a delicate challenge for our future works. The benefit of such a method could simplify the process of numerical computation.

**Keywords:** boundary value problems for non-linear parabolic PDE; fractional step method; convergence of numerical methods; numerical algorithm; error analysis; dynamic boundary conditions

**MSC:** 35K55; 65N06; 65N12; 65YXX; 80AXX

## 1. Introduction

Considering the following non-linear second-order reaction–diffusion problem:

$$
\begin{cases}
p_1 \dfrac{\partial}{\partial t} U(t,x) - p_2 \mathrm{div}\Big(K\big(t,x,U(t,x)\big)\,\nabla U(t,x)\Big) \\
\qquad = p_r\Big[U(t,x) - U^3(t,x)\Big] + p_s g_d(t,x) & \text{in } Q \\
p_2 \dfrac{\partial}{\partial \mathbf{n}} U + p_1 \dfrac{\partial}{\partial t} U - \Delta_\Gamma U + p_t U = g_{fr}(t,x) & \text{on } \Sigma \\
U(0,x) = U_0(x) & \text{on } \Omega,
\end{cases}
\tag{1}
$$

where $\Omega \subset \mathbf{R}^n$, $n \leq 3$ is a compact domain with a $C^2$ boundary $\partial\Omega = \Gamma$, $[0,T]$ a generic time interval, $Q = (0,T] \times \Omega$, $\Sigma = (0,T] \times \partial\Omega$ and:

- $t \in (0,T]$, $x = (x_1, \dots, x_n) \in \Omega$;
- $p_1$, $p_2$, $p_r$, $p_s$ and $p_t$ are positive parameters;

- $\dfrac{\partial}{\partial s}U(s,\cdot)$ ($U_s$ in short) is the partial derivative of $U(s,\cdot)$ ($U$ in short) relative to $s \in (0,T]$;

- $U(s,y)$, $(s,y) \in Q$, is the unknown function (the order parameter in $Q$, for example). $\nabla U(s,y) = U_y(s,y)$ ($\nabla U = U_y$) denotes the gradient of $U(s,y)$ in $y$, $y \in \Omega$ (see [1–3] for more details);

- $K\big(s,y,U(s,y)\big)$ is the mobility (attached to the solution $U(s,y)$, $(s,y) \in Q$, to Equation (1)) (see [2–4] for more details);

- $g_d(s,y) \in L^p(Q)$ is the distributed control (see Remark 1 below), where

$$p \geq 2; \tag{2}$$

- $g_{fr}(s,y) \in W_p^{1-\frac{1}{2p},2-\frac{1}{p}}(\Sigma)$ is the boundary control (see Remark 1 below);

- $U_0 \in W_\infty^{2-\frac{2}{p}}(\Omega)$ verifying

$$p_2\frac{\partial}{\partial n}U_0 - \Delta_\Gamma U_0 + p_t U_0 = g_{fr}(0,x);$$

- $\mathbf{n} = n(x)$ has the same meaning as in [5];

- $\Delta_\Gamma$ has the same meaning as in [6];

**Remark 1.** *The given functions $g_d$ and $g_{fr}$ in (1), can be interpreted as distributed and boundary control, respectively, opening a large field of applications for the non-linear second-order problem (1), such as optimal control.*

For convenience, let us write (1) in the following form

$$\begin{cases} p_1\dfrac{\partial}{\partial t}U(t,x) - p_2\dfrac{\partial}{\partial U_{x_j}}\Big[K(t,x,U(t,x))U_{x_i}\Big]U_{x_j x_i} \\[2mm] \qquad = A\Big(t,x,U(t,x),U_{x_i}(t,x)\Big) + p_r\Big[U(t,x) - U^3(t,x)\Big] + p_s g_d(t,x) \quad \text{in } Q \\[2mm] p_2\dfrac{\partial}{\partial \mathbf{n}}U + p_1\dfrac{\partial}{\partial t}U - \Delta_\Gamma U + p_t U = g_{fr}(t,x) \qquad\qquad\qquad\qquad \text{on } \Sigma \\[2mm] U(0,x) = U_0(x) \qquad\qquad\qquad\qquad\qquad\qquad\qquad\qquad\qquad \text{on } \Omega, \end{cases} \tag{3}$$

where $U_{x_j x_i} = \dfrac{\partial^2}{\partial x_j \partial x_i}U(t,x)$, $i,j = 1,\ldots,n$, and

$$A\big(t,x,U(t,x),U_{x_i}(t,x)\big) = \frac{\partial}{\partial U}(K(t,x,U)U_{x_i})U_{x_i} + \frac{\partial}{\partial x_i}(K(t,x,U)U_{x_i}),\ i=1,\ldots,n. \tag{4}$$

As in [1–3,5–9], we recall that Equation (1)$_1$ is a quasi-linear one, i.e.,

$$a_i(t,x,U(t,x),U_x(t,x)) = K(t,x,U(t,x))U_{x_i}(t,x),\quad i=1,\ldots,n$$

and

$$a(t,x,U(t,x),U_x(t,x)) = -p_r\Big[U(t,x) - U^3(t,x)\Big] - p_s g_d(t,x).$$

On the other hand, the problem in (3)$_1$ is similar to in [10] (p. 3, relation (2.4)), where, for $i = 1,\ldots,n$,

$$a_{ij}\big(t,x,U(t,x),U_x(t,x)\big) = \frac{\partial}{\partial U_{x_j}}a_i(t,x,U(t,x),U_x(t,x)) = \frac{\partial}{\partial U_{x_j}}\Big[K(t,x,U(t,x))U_{x_i}(t,x)\Big],$$

and

$$a(t, x, U(t, x), U_x(t, x)) = -A(t, x, U(t, x), U_x(t, x)) - p_r \left[ U(t, x) - U^3(t, x) \right] - p_s g_d(t, x),$$

while $(3)_2$ are of the second type, namely

$$\frac{\partial}{\partial \mathbf{n}} U(t, x) = a_{ij}(t, x, U(t, x), U_x(t, x)) U_{x_j}(t, x) \cos \alpha_i,$$

and

$$\psi(t, x, U)|_\Sigma = p_1 \frac{\partial}{\partial t} U - \Delta_\Gamma U + p_t U - g_{fr}(t, x) \tag{5}$$

(see [10] (p. 475, relation (7.2))).

Moreover, we consider that Equations $(1)_1$ and $(3)_1$ are uniformly parabolic, i.e.,

$$\nu_1(|U|)\zeta^2 \le \frac{\partial}{\partial z_j} a_i(s, y, U(s, y), z(s, y))\zeta_i \zeta_j \le \nu_2(|U|)\zeta^2 \tag{6}$$

for arbitrary $U(s, y)$ and $z(s, y)$, $(s, y) \in Q$, and $\zeta = (\zeta_1, \ldots, \zeta_n)$ for an arbitrary real vector (see [5] for more details).

Equation $(1)_1$ was initially introduced by Allen and Cahn (see [5,11] and references therein) to describe the motion of anti-phase boundaries in crystalline solids. In fact, the Allen–Cahn model is widely applied to moving interface problems, such as the mixture of two incompressible fluids, the nucleation of solids, vesicle membranes, etc. Furthermore, the non-linear parabolic Equation $(1)_1$ appears in the Caginalp's phase-field transition system (see [2–9,11–22]), describing the transition between phases (solid and liquid) (see [17], for example).

In the present paper we investigate the solvability of boundary value problems of the form (1) or (3) in the class $W_p^{1,2}(Q)$. The new model expressed in (1) stands out by the presence of parameters $p_1$, $p_2$, $p_r$, $p_s$, $p_t$, $K(s, y, U(s, y))$, and $(s, y) \in Q$, the principal part being in the divergence form and by considering a non-linear reaction term (see [5,11] and references therein). The most important aspect in our paper concerns inhomogeneous dynamic boundary conditions. Thus, we more precisely define the significant aspects of the physical features. In this regard, we advise applying (1) or (3), to the moving interface problems (see [5,7,8,11–15]), anisotropy effects (see [3–6,9,11,16–22]), image de-noising and segmentation (see [2,4] and references therein), etc. Let us point out that the following assumption is satisfied (see [20]):

$$H_0: \quad (U - U^3)|U|^{3p-4}U \le 1 + |U|^{3p-1} - |U|^{3p}.$$

## 2. Results—Theorem 1

In order to approach the problem in (3) (or (1)), we use the same ideas as in [1,6,7,9]. In this respect we introduce a new variable $\zeta(t, x) = U(t, x)$, $\zeta(0, x) = U_0(x)$ on $\Gamma$ (see [10] (6.2)). Correspondingly, $(3)_2$ is approached in the following

$$\begin{cases} U(t, x) = \zeta(t, x) & \text{on } \Sigma \\[2mm] p_2 \frac{\partial}{\partial \mathbf{n}} U + p_1 \frac{\partial}{\partial t} \zeta(t, x) - \Delta_\Gamma \zeta(t, x) + p_t \zeta(t, x) = g_{fr}(t, x) & \text{on } \Sigma \\[2mm] \zeta(0, x) = \zeta_0(x) & x \in \Gamma. \end{cases} \tag{7}$$

Accordingly, the non-linear second-order boundary value problem (3) can be written suitably as follows

$$
\begin{cases}
p_1 \dfrac{\partial}{\partial t} U(t,x) - p_2 \dfrac{\partial}{\partial U_{x_j}} \Big[ K(t,x,U(t,x)) U_{x_i}(t,x) \Big] U_{x_j x_i} \\[2mm]
\qquad = A\Big(t,x,U(t,x),U_{x_i}(t,x)\Big) + p_r \Big[ U(t,x) - U^3(t,x) \Big] + p_s g_d(t,x) & \text{in } Q \\[2mm]
U(t,x) = \zeta(t,x) & \text{on } \Sigma \\[2mm]
p_2 \dfrac{\partial}{\partial \mathbf{n}} U + p_1 \dfrac{\partial}{\partial t} \zeta - \Delta_\Gamma \zeta + p_t \zeta = g_{fr}(t,x) & \text{on } \Sigma \\[2mm]
U(0,x) = U_0(x) & \text{on } \Omega \\[2mm]
\zeta(0,x) = \zeta_0(x) & x \in \Gamma,
\end{cases}
\tag{8}
$$

where $A\big(t,x,U(t,x),U_{x_i}(t,x)\big)$ is defined by (4), $U_0(x) = \zeta_0(x)$ on $\Gamma$ and $\zeta_0(x) \in W_\infty^{2-\frac{2}{p}}(\Gamma)$.

**Definition 1.** *Any solution $\big(U(t,x),\zeta(t,x)\big)$ to problem (8) is called the* `classical solution` *if it is continuous in $\bar{Q}$, with continuous derivatives $U_t$, $U_x$ and $U_{xx}$ in $Q$ and $\zeta_t$, $\zeta_x$, and $\zeta_{xx}$ in $\Sigma$, satisfying Equation (8)$_1$ at all points $(t,x) \in Q$ and satisfying conditions (8)$_{2-3}$ and (8)$_{4-5}$ on the lateral surface $\Sigma$ of the cylinder $Q$ for $t = 0$, respectively.*

Our main results regarding the existence, uniqueness and regularity of solutions to problem (8) (the well-posedness of the solutions to the non-linear second-order boundary value problems (1) or (3)) are presented below.

**Theorem 1.** *Suppose $\big(U(t,x),\zeta(t,x)\big) \in C^{1,2}(Q) \times C^{1,2}(\Sigma)$ is a classical solution to problem (8), and for positive numbers $M$, $M_0$, $m_1$, $M_1$, $M_2$, $M_3$, $M_4$ and $M_5$ one has*

**I$_1$.**  *$|U(t,x)| < M$ for any $(t,x) \in Q$ and for any $z(t,x)$, the map $K(t,x,z)$ is continuous, differentiable in $x$, where its $x$-derivatives are bounded, satisfy (6), and*

$$
0 < K_{min} \leq K\big(t,x,U(t,x)\big) < K_{max}, \quad for \ (t,x) \in Q,
\tag{9}
$$

$$
\sum_{i=1}^{n} \left[ \big|a_i(t,x,U(t,x),z(t,x))\big| + \left| \frac{\partial}{\partial U} a_i(t,x,U(t,x),z(t,x)) \right| \right] (1+|z|)
$$
$$
+ \sum_{i,j=1}^{n} \left| \frac{\partial}{\partial x_j} a_i(t,x,U(t,x),z(t,x)) \right| + |U(t,x)| \leq M_0 (1+|z|)^2.
\tag{10}
$$

**I$_2$.**  *For any sufficiently small $\varepsilon > 0$, the functions $U(t,x)$ and $K(t,x,U(t,x))$ satisfy the relations*

$$
\|U\|_{L^s(Q)} \leq M_2, \qquad \|K(t,x,U(t,x))U_{x_i}\|_{L^r(Q)} < M_3, \ i = 1,\ldots,n,
$$

*where*

$$
r = \begin{cases} \max\{p,4\} & p \neq 4 \\ 4+\varepsilon & p = 4, \end{cases} \qquad
s = \begin{cases} \max\{p,2\} & p \neq 2 \\ 2+\varepsilon & p = 2. \end{cases}
$$

*Then, when $\forall g_d \in L^p(Q)$, $U_0 \in W_\infty^{2-\frac{2}{p}}(\Omega)$, $\zeta_0(x) \in W_\infty^{2-\frac{2}{p}}(\Gamma)$, $g_{fr} \in W_p^{1-\frac{1}{2p},2-\frac{1}{p}}(\Sigma)$, with $p \neq \frac{3}{2}$, there exists a unique solution $(U,\zeta) \in W_p^{1,2}(Q) \times W_p^{1,2}(\Sigma)$ to (8) which satisfies*

$$\|U\|_{W_p^{1,2}(Q)} + \|\zeta\|_{W_p^{1,2}(\Sigma)}$$

$$\leq C\left\{1 + \|U_0\|_{W_\infty^{2-\frac{2}{p}}(\Omega)} + \|\zeta_0\|_{W_\infty^{2-\frac{2}{p}}(\Gamma)} + \|U_0\|_{L^{3p-2}(\Omega)}^{\frac{3p-2}{p}} + \|\zeta_0\|_{L^{3p-2}(\Gamma)}^{\frac{3p-2}{p}}\right.$$

$$\left. + \|g_d\|_{L^{3p-2}(Q)}^{\frac{3p-2}{p}} + \|g_{fr}\|_{L^{3p-2}(\Sigma)}^{\frac{3p-2}{p}} + \|g_{fr}\|_{W_p^{1-\frac{1}{2p},2-\frac{1}{p}}(\Sigma)}\right\}, \tag{11}$$

*where $C > 0$ does not depend on $U$, $\zeta$, $g_d$, or $g_{fr}$.*

*If $(U^1, \zeta^1)$ and $(U^2, \zeta^2)$ are solutions to (8) which correspond to $(U_0^1, \zeta_0^1)$, $(U_0^2, \zeta_0^2) \in W_\infty^{2-\frac{2}{p}}(\Omega) \times W_\infty^{2-\frac{2}{p}}(\Gamma)$, $g_d^1$, $g_d^2$, $g_{fr}^1$ and $g_{fr}^2$, respectively, then*

$$\|U^1\|_{W_p^{1,2}(Q)}, \quad \|U^2\|_{W_p^{1,2}(Q)} \leq M_4, \tag{12}$$

$$\|\zeta^1\|_{W_p^{1,2}(\Sigma)}, \quad \|\zeta^2\|_{W_p^{1,2}(\Sigma)} \leq M_5, \tag{13}$$

*and the following holds*

$$\max_{(t,x)\in Q} |U^1 - U^2| + \max_{(t,x)\in\Sigma} |\zeta^1 - \zeta^2|$$

$$\leq C_1 e^{CT} \max\left\{\max_{(t,x)\in\Omega}|U_0^1 - U_0^2|, \max_{(t,x)\in\Gamma}|\zeta_0^1 - \zeta_0^2|,\right.$$

$$\left.\max_{(t,x)\in Q}|g_d^1 - g_d^2|, \max_{(t,x)\in\Sigma}|g_{fr}^1 - g_{fr}^2|\right\}, \tag{14}$$

*where $C_1 > 0$ and $C > 0$, do not depend on $\left\{U^1, \zeta^1, g_d^1, g_{fr}^1, U_0^1, \zeta_0^1\right\}$ and $\left\{U^2, \zeta^2, g_d^2, g_{fr}^2, U_0^2, \zeta_0^2\right\}$. In particular, the uniqueness of the solution to (8) holds.*

As far as the techniques used in this paper are concerned, it should be noted that we derive the a priori estimates for $L^p(Q)$ and $L^p(\Sigma)$. Moreover, basic tools in our approach are:

- the Leray–Schauder degree theory (see [15] (p. 221) and reference therein);
- the $L^p$ theory of linear and quasi-linear parabolic equations [10];
- Green's first identity

$$-\int_\Omega y \operatorname{div} z \, dx = \int_\Omega \nabla y \cdot z \, dx - \int_{\partial\Omega} y \frac{\partial}{\partial \mathbf{n}} z \, d\gamma,$$

$$-\int_\Omega y \Delta z \, dx = \int_\Omega \nabla y \cdot \nabla z \, dx - \int_{\partial\Omega} y \frac{\partial}{\partial \mathbf{n}} z \, d\gamma, \tag{15}$$

  for any scalar-valued function $y$ and $z$ in a continuously differentiable vector field in $n$ dimensional space;

- the Lions and Peetre embedding theorem [1] (p. 100) to ensure the existence of a continuous embedding $W_p^{1,2}(Q) \subset L^\mu(Q)$, where the number $\mu$ is defined as follows (see (2))

$$\mu = \begin{cases} \text{any positive number } \geq 3p & \text{if } \dfrac{1}{p} - \dfrac{2}{n+2} \leq 0, \\[3mm] \dfrac{p(n+2)}{n+2-2p} & \text{if } \dfrac{1}{p} - \dfrac{2}{n+2} > 0. \end{cases} \tag{16}$$

For a given positive integer $k$ and $1 \leq p \leq \infty$, we denote by $W_p^{k,2k}(Q)$ the Sobolev space on $Q$:

$$W_p^{k,2k}(Q) = \left\{ y \in L^p(Q) : \frac{\partial^i}{\partial t^i}\frac{\partial^j}{\partial x^j}\, y \in L^p(Q), \text{ for } 2i + j \leq 2k \right\},$$

i.e., the spaces of functions whose $t$- and $x$-derivatives up to the order $k$ and $2k$, respectively, belong to $L^p(Q)$. Furthermore, we use the Sobolev spaces $W_p^i(\Omega)$ and $W_p^{\frac{i}{2},i}(\Sigma)$ with the non-integral $i$ for the initial and boundary conditions, respectively, (see [10] (p. 70 and 81)).

Furthermore, we use the set $C^{1,2}(\bar{D})$ ($C^{1,2}(D)$) of all continuous functions in $\bar{D}$ (in $D$) with continuous derivatives $u_t$, $u_x$, and $u_{xx}$ in $\bar{D}$ (in $D$) ($D = Q$ or $D = \Sigma$), as well as the Sobolev spaces $W_p^\ell(\Omega)$, and $W_p^{\ell,\ell/2}(\Sigma)$ with non-integral $\ell$ for the initial and boundary conditions, respectively (see [10] (p. 8, p. 70 and p. 81)).

In the following we will denote by $C$ some positive constants.

### 3. Proof of the Main Result — Theorem 1

We consider $B = W_p^{0,1}(Q) \cap L^{3p}(Q) \times L^p(\Sigma)$ as a suitable Banach space, with the norm $\|\cdot\|_B$ expressed by

$$\|(\varphi, \bar{\varphi})\|_B = \|\varphi\|_{L^p(Q)} + \|\varphi_x\|_{L^p(Q)} + \|\bar{\varphi}\|_{L^p(\Sigma)},$$

and a non-linear operator $H : B \times [0,1] \to B$ defined by

$$(U,\zeta) = H(\varphi, \bar{\varphi}, \lambda) = \big(U(\varphi, \bar{\varphi}, \lambda), \zeta(\varphi, \bar{\varphi}, \lambda)\big) \quad \forall (\varphi, \bar{\varphi}) \in B, \ \forall \lambda \in [0,1], \tag{17}$$

where $(U(\varphi, \bar{\varphi}, \lambda), \zeta(\varphi, \bar{\varphi}, \lambda)$ is a unique solution to the following linear second-order boundary value problem

$$\begin{cases} p_1 \dfrac{\partial}{\partial t} U - p_2 \left[ \lambda \dfrac{\partial}{\partial \varphi_{x_j}}(K(t,x,\varphi)\varphi_{x_i}) - (1-\lambda)\delta_i^j \right] U_{x_i x_j} \\[2mm] \qquad\qquad = \lambda \left\{ A\big(t,x,\varphi,\varphi_{x_i}\big) + p_r \big[\varphi(t,x) - \varphi^3(t,x)\big] + p_s g_d(t,x) \right\} \quad \text{in } Q \\[2mm] U(t,x) = \zeta(t,x) \qquad\qquad\qquad\qquad\qquad\qquad\qquad\qquad\qquad\quad \text{on } \Sigma \\[2mm] U(0,x) = \lambda U_0(x) \qquad\qquad\qquad\qquad\qquad\qquad\qquad\qquad\qquad \text{on } \Omega \\[2mm] p_2 \dfrac{\partial}{\partial \mathbf{n}} U + p_1 \dfrac{\partial}{\partial t}\zeta - \Delta_\Gamma \zeta + p_t \zeta = \lambda g_{fr}(t,x) \qquad\qquad\qquad \text{on } \Sigma \\[2mm] \zeta(0,x) = \lambda \zeta_0(x) \qquad\qquad\qquad\qquad\qquad\qquad\qquad\qquad\qquad\quad x \in \Gamma. \end{cases} \tag{18}$$

**Remark 2.** *The non-linear operator $H$ in (17) depends on $\lambda \in [0,1]$ and its fixed point for $\lambda = 1$ is a solution to problem (18).*

**Proof.** We now prove that the non-linear operator $H$, defined in (17), is well-defined, continuous and compact.

From the right-hand side of (17)$_1$, it follows that, $\forall (\varphi, \bar{\varphi}) \in B$, then $\varphi^3 \in L^p(Q)$ and thus $A\big(t,x,\varphi,\varphi_{x_i}\big) + p_r \big[\varphi(t,x) - \varphi^3(t,x)\big] + p_s g_d(t,x) \in L^p(Q)$. Using the $L^p$ theory of linear parabolic equations (see [10]), the solution $(U,\zeta)$ to problem (18) exists and it is unique with

$$(U,\zeta) = \big(U(\varphi, \bar{\varphi}, \lambda), \zeta(\varphi, \bar{\varphi}, \lambda)\big) \in B, \quad \forall (\varphi, \bar{\varphi}) \in B, \ \forall \lambda \in [0,1]. \tag{19}$$

Using the continuous inclusions (see [6])

$$
\begin{cases}
W_p^{1,2}(Q) \subset B \subset L^p(Q) \\[2mm]
W_p^{1,2}(\Sigma) \subset L^p(\Sigma),
\end{cases}
\tag{20}
$$

we obtain $H(\varphi, \bar{\varphi}, \lambda) = (U, \zeta) \in B$ for all $(\varphi, \bar{\varphi}) \in B$ and $\forall\, \lambda \in [0, 1]$, meaning the non-linear operator $H$ is well defined.

Now, using the ideas from [1–7,9,16,20], let $\varphi^n \to \varphi$ in $W_p^{0,1}(Q) \cap L^{3p}(Q)$, $\bar{\varphi}^n \to \bar{\varphi}$ in $L^p(\Sigma)$ and $\lambda^n \to \lambda$ in $[0, 1]$. Using the notations

$$
\begin{aligned}
(U^{n,\lambda_n}, \zeta^{n,\lambda_n}) &= H(\varphi^n, \bar{\varphi}^n, \lambda^n), \\
(U^{n,\lambda}, \zeta^{n,\lambda}) &= H(\varphi^n, \bar{\varphi}^n, \lambda), \\
(U^{\lambda}, \zeta^{\lambda}) &= H(\varphi, \bar{\varphi}, \lambda),
\end{aligned}
$$

we obtain

$$
\|u^{n,\lambda_n} - u^{n,\lambda}\|_{W_p^{1,2}(Q)} + \|\zeta^{n,\lambda_n} - \zeta^{n,\lambda}\|_{W_p^{1,2}(\Sigma)} \to 0 \quad \text{for } n \to \infty
\tag{21}
$$

and

$$
\|u^{n,\lambda} - u^{\lambda}\|_{W_p^{1,2}(Q)} + \|\zeta^{n,\lambda} - \zeta^{\lambda}\|_{W_p^{1,2}(\Sigma)} \to 0 \quad \text{for } n \to \infty.
\tag{22}
$$

The continuous embedding of (20), (21), and (22) allows us to derive the continuity of the non-linear operator $H$, introduced in (17). Furthermore, $H$ is compact, easily written as

$$
B \times [0, 1] \to W_p^{1,2}(Q) \times W_p^{1,2}(\Sigma) \hookrightarrow B = W_p^{0,1}(Q) \cap L^{3p}(Q) \times L^p(\Sigma),
$$

where the second map is a compact inclusion (see [1] (p. 100)).

Next, we look at a positive number $R$, such that (see (17))

$$
(U, \zeta, \lambda) \in B \times [0, 1] \quad \text{with } (U, \zeta) = H(U, \zeta, \lambda) \implies \|(U, \zeta)\|_B < R.
\tag{23}
$$

The above expression $(U, \zeta) = H(U, \zeta, \lambda)$ can be written as (see (1), (8) and (18))

$$
\begin{cases}
p_1 \dfrac{\partial}{\partial t} U - \lambda p_2 \mathrm{div}\Big(K(t, x, U)\nabla U\Big) - (1 - \lambda)p_2 \Delta U \\[4mm]
\qquad\qquad = \lambda\Big[p_r\big[U(t, x) - U^3(t, x)\big] + p_s g_d(t, x)\Big] & \text{in } Q \\[4mm]
U(t, x) = \zeta(t, x) & \text{on } \Sigma \\[3mm]
U(0, x) = \lambda U_0(x) & \text{on } \Omega \\[3mm]
p_2 \dfrac{\partial}{\partial \mathbf{n}} U + p_1 \dfrac{\partial}{\partial t}\zeta - \Delta_\Gamma \zeta + p_t \zeta = \lambda g_{fr}(t, x)] & \text{on } \Sigma \\[4mm]
\zeta(0, x) = \lambda \zeta_0(x) & x \in \Gamma.
\end{cases}
\tag{24}
$$

Multiplying $(24)_1$ by $|U|^{3p-4}U$ and integrating over $Q_s := (0,s) \times \Omega,\ s \in (0,T]$, we obtain

$$
\begin{aligned}
&\frac{p_1}{3p-2} \int_\Omega |U(s,x)|^{3p-2}\, dx \\
&\quad -\lambda p_2 \int_{Q_s} \mathrm{div}\left(K(\tau,x,U)\nabla U\right)|U|^{3p-4}U\, d\tau dx \\
&\quad -(1-\lambda)p_2 \int_{Q_s} \Delta U\, |U|^{3p-4}U\, d\tau dx \\
&= \lambda p_r \int_{Q_s} \left[U(\tau,x) - U^3(\tau,x)\right]|U|^{3p-4}U\, d\tau dx + \lambda p_s \int_{Q_s} g_d(\tau,x)|U|^{3p-4}U\, d\tau dx.
\end{aligned}
\tag{25}
$$

To process the terms

$$
\int_{Q_s} \mathrm{div}\left(K(\tau,x,U)\nabla U\right)|U|^{3p-4}U d\tau dx
$$

and

$$
\int_{Q_s} \Delta U\, |U|^{3p-4}U d\tau dx, \text{ in } (25)
$$

we use Green's first identity $(15)_1$ and $(15)_2$, respectively, to obtain

$$
\begin{aligned}
&-\lambda p_2 \int_{Q_s} \mathrm{div}\left(K(\tau,x,U)\nabla U\right)|U|^{3p-4}U\, d\tau dx \\
&= \lambda p_2 \int_{Q_s} K(\tau,x,U)\nabla U \cdot \nabla\left(|U|^{3p-4}U\right) d\tau dx + \lambda \int_{\Sigma_s} |U|^{3p-4}U \left(-p_2 \frac{\partial}{\partial \mathbf{n}}U\right) d\tau d\gamma,
\end{aligned}
\tag{26}
$$

$$
\begin{aligned}
&-(1-\lambda)p_2 \int_{Q_s} \Delta U\, |U|^{3p-4}U\, d\tau dx \\
&= (1-\lambda)3(p-1)p_2 \int_{Q_s} |\nabla U|^2 |U|^{3p-4} d\tau dx + (1-\lambda)\int_{\Sigma_s} |U|^{3p-4}U\left(-p_2\frac{\partial}{\partial \mathbf{n}}U\right) d\tau d\gamma,
\end{aligned}
\tag{27}
$$

where $\Sigma_s = (0,s) \times \partial\Omega,\ s \in (0,T]$ and

$$
-p_2 \frac{\partial}{\partial \mathbf{n}}U = p_1 \frac{\partial}{\partial t}\zeta - \Delta_\Gamma \zeta + p_t \zeta - \lambda g_{fr}
$$

(see $(24)_4$).

Combining the above equality with the boundary condition in $(24)_2$, the left inequality in (9), and the relations (26), (27), and (25) leads us to the following inequality

$$\frac{p_1}{3p-2}\int_\Omega |U(s,x)|^{3p-2}dx+\lambda\frac{p_1}{3p-2}\int_\Gamma |\zeta(s,x)|^{3p-2}d\gamma+(1-\lambda)\frac{p_1}{3p-2}\int_\Gamma |\zeta(s,x)|^{3p-2}d\gamma$$

$$+\lambda p_2\int_{Q_s}K(\tau,x,U)\nabla U\cdot\nabla\left(|U|^{3p-4}U\right)d\tau dx+(1-\lambda)3(p-1)p_2\int_{Q_s}|\nabla U|^2|U|^{3p-4}d\tau dx$$

$$+\lambda p_t\int_{\Sigma_s}|\zeta(\tau,x)|^{3p-2}\,d\tau d\gamma+(1-\lambda)p_t\int_{\Sigma_s}|\zeta(\tau,x)|^{3p-2}\,d\tau d\gamma$$

$$+\lambda\int_{\Sigma_s}\nabla_\Gamma\left(|\zeta|^{3p-3}\right)\cdot\nabla_\Gamma\zeta\,d\tau d\gamma+(1-\lambda)\int_{\Sigma_s}\nabla_\Gamma\left(|\zeta|^{3p-3}\right)\cdot\nabla_\Gamma\zeta\,d\tau d\gamma \qquad (28)$$

$$\le\lambda\frac{p_1}{3p-2}\int_\Omega |U_0(x)|^{3p-2}\,dx+\frac{p_1}{3p-2}\int_\Gamma |\zeta_0(x)|^{3p-2}d\gamma$$

$$+\lambda p_r\int_{Q_s}\left[U(\tau,x)-U^3(\tau,x)\right]|U|^{3p-4}U\,d\tau dx$$

$$+\lambda p_s\int_{Q_s}g_d(\tau,x)|U|^{3p-4}U\,d\tau dx+\lambda\int_{\Sigma_t}g_{f^r}(\tau,x)|U|^{3p-4}U\,d\tau d\gamma$$

for all $s\in(0,T]$. The last two terms in the above inequalities can be manipulated via Hölder and Cauchy's inequality giving us the following estimates

a. $\lambda p_s\displaystyle\int_{Q_s}g_d(\tau,x)|U|^{3p-4}Ud\tau dx$

$$\le\frac{(3p-2)-1}{3p-2}\varepsilon^{\frac{3p-2}{3p-3}}\int_{Q_s}|U|^{3p-2}d\tau dx+\lambda p_s\frac{1}{3p-2}\varepsilon^{-(3p-2)}\int_{Q_s}|g_d|^{3p-2}d\tau dx,$$

b. $\lambda\displaystyle\int_{\Sigma_s}g_{f^r}(\tau,x)|U|^{3p-4}Ud\tau d\gamma$

$$\le\frac{(3p-2)-1}{3p-2}\varepsilon^{\frac{3p-2}{3p-3}}\int_{\Sigma_s}|U|^{3p-2}d\tau d\gamma+\lambda\frac{1}{3p-2}\varepsilon^{-(3p-2)}\int_{\Sigma_t}|g_{f^r}|^{3p-2}d\tau d\gamma.$$

Due to the inequalities **a.** and **b.**, from (28) we obtain

$$
\frac{p_1}{3p-2}\left[\int_\Omega |U(s,x)|^{3p-2}dx + \int_\Gamma |\zeta(s,x)|^{3p-2}d\gamma\right]
$$

$$
+\lambda p_2 \int_{Q_s} K(\tau,x,U)\nabla U \cdot \nabla\left(|U|^{3p-4}U\right)d\tau dx + (1-\lambda)3(p-1)p_2\int_{Q_s}|\nabla U|^2|U|^{3p-4}d\tau dx
$$

$$
+\lambda p_r \int_{Q_s} |U(\tau,x)|^{3p}\, d\tau dx
$$

$$
+p_t \int_{\Sigma_s} |\zeta(\tau,x)|^{3p-2}\, d\tau d\gamma + \int_{\Sigma_s} \nabla_\Gamma\left(|\zeta|^{3p-3}\right)\cdot\nabla_\Gamma \zeta \, d\tau d\gamma
$$

$$
\leq \frac{p_1}{3p-2}\left[\int_\Omega |U_0(x)|^{3p-2}\, dx + \int_\Gamma |\zeta_0(x)|^{3p-2}d\gamma\right]
$$

$$
+\left[\lambda p_r + \frac{(3p-2)-1}{3p-2}\varepsilon^{\frac{3p-2}{3p-3}}\right]\int_{Q_s} |U(\tau,x)|^{3p-2}d\tau dx
$$

$$
+\frac{(3p-2)-1}{3p-2}\varepsilon^{\frac{3p-2}{3p-3}}\int_{\Sigma_s} |U(\tau,x)|^{3p-2}\, d\tau dx
$$

$$
+p_s \frac{1}{3p-2}\varepsilon^{-(3p-2)}\|g_d\|^{3p-2}_{L^{3p-2}(Q_s)} + \frac{1}{3p-2}\varepsilon^{-(3p-2)}\|g_{f^r}\|^{3p-2}_{L^{3p-2}(\Sigma_s)}
$$

(29)

for all $s \in (0,T]$.

In particular, it follows that from (29) we obtain

$$
\int_\Omega |U(s,x)|^{3p-2}dx + \int_\Gamma |\zeta(s,x)|^{3p-2}d\gamma
$$

$$
\leq C_0\left[\|U_0(x)\|^{3p-2}_{L^{3p-2}(\Omega)} + \|\zeta_0(x)\|^{3p-2}_{L^{3p-2}(\Gamma)} + \|g_d\|^{3p-2}_{L^{3p-2}(Q_s)} + \|g_{f^r}\|^{3p-2}_{L^{3p-2}(\Sigma_s)}\right]
$$

$$
+C_0\int_0^t\left[\int_\Omega |U(\tau,x)|^{3p-2}d\tau dx + \int_\Gamma |\zeta(\tau,x)|^{3p-2}d\gamma\right]d\tau
$$

(30)

where $C_0 = C(|\Omega|, |\Gamma|, p, p_1, p_2, p_r, p_t, p_s)$, in conjuction with (24)$_2$.

By Gronwall's lemma and owing to $L^{3p-2}(Q) \subset L^p(Q)$, from (30) we obtain

$$
\|U\|^p_{L^p(Q)} + \|\zeta\|^p_{L^p(\Sigma)}
$$

$$
\leq C(T,C_0)\left[\|U\|^{3p-2}_{L^{3p-2}(Q)} + \|\zeta\|^{3p-2}_{L^{3p-2}(\Sigma)}\right]
$$

(31)

$$
\leq C(T,C_0)\left[\|U_0(x)\|^{3p-2}_{L^{3p-2}(\Omega)} + \|\zeta_0(x)\|^{3p-2}_{L^{3p-2}(\Gamma)} + \|g_d\|^{3p-2}_{L^{3p-2}(Q)} + \|g_{f^r}\|^{3p-2}_{L^{3p-2}(\Sigma)}\right].
$$

Having established an estimate for $\|U\|^{3p-2}_{L^{3p-2}(Q)} + \|\zeta\|^{3p-2}_{L^{3p-2}(\Sigma)}$ (see (31)), we now return to the relation in (29) to derive the following estimate:

$$\lambda p_r \||U|^3\|^p_{L^p(Q)}$$

$$\leq C(T, C_0)\left[\|U_0(x)\|^{3p-2}_{L^{3p-2}(\Omega)} + \|\zeta_0(x)\|^{3p-2}_{L^{3p-2}(\Gamma)} + \|g_d\|^{3p-2}_{L^{3p-2}(Q)} + \|g_{f^r}\|^{3p-2}_{L^{3p-2}(\Sigma)}\right], \tag{32}$$

where the boundary condition in $(24)_2$ is also used.

Applying Lemma 7.4 in Choban and Moroşanu [1] (p. 114) to the linear inhomogeneous problem (24) with

$$f_3 = \lambda\{p_r[U(t,x) - U^3(t,x)] + p_s g_d(t,x)\} \in L^p(Q) \text{ and}$$

$$g_3 = \lambda g_{fr}(t,x) \in L^p(\Sigma),$$

we obtain

$$\|U\|_{W_p^{1,2}(Q)} + \|\zeta\|_{W_p^{1,2}(\Sigma)}$$

$$\leq C_1\Bigg\{\|U_0\|_{W_\infty^{2-\frac{2}{p}}(\Omega)} + \|\zeta_0\|_{W_\infty^{2-\frac{2}{p}}(\Gamma)} + \|g_d\|_{L^p(Q)} + \|g_{fr}\|_{L^p(\Sigma)} \tag{33}$$

$$+ \lambda p_r\left[\|U\|_{L^p(\Omega)} + \||U|^3\|_{L^p(\Omega)}\right]\Bigg\},$$

for a constant $C_1 = C(n, C(T, C_0)) > 0$.

Now using (31) and (32), (33) then becomes

$$\|U\|_{W_p^{1,2}(Q)} + \|\zeta\|_{W_p^{1,2}(\Sigma)}$$

$$\leq C_1\Bigg\{1 + \|U_0\|_{W_\infty^{2-\frac{2}{p}}(\Omega)} + \|\zeta_0\|_{W_\infty^{2-\frac{2}{p}}(\Gamma)} + \|U_0\|^{\frac{3p-2}{p}}_{L^{3p-2}(\Omega)} + \|\zeta_0\|^{\frac{3p-2}{p}}_{L^{3p-2}(\Gamma)} \tag{34}$$

$$+ \|g_d\|^{\frac{3p-2}{p}}_{L^{3p-2}(Q)} + \|g_{fr}\|^{\frac{3p-2}{p}}_{L^{3p-2}(\Sigma)} + \|g_d\|_{L^p(Q)} + \|g_{fr}\|_{L^p(\Sigma)}\Bigg\},$$

The inclusions in (20) guarantee that

$$\|U\|_{L^p(Q)} + \|\zeta\|_{L^p(\Sigma)} \leq C\left(\|U\|_{W_p^{1,2}(Q)} + \|\zeta\|_{W_p^{1,2}(\Sigma)}\right)$$

where, thanks to (34), we may conclude that a constant $R > 0$ exists such that the property in (23) is true.

Denoting $B_R^H := \{(U, \zeta) \in B : \|(U, \zeta)\|_B < R\}$, relation (23) implies that

$$(U, \zeta, \lambda) \neq (U, \zeta) \quad \forall (U, \zeta) \in \partial B_R^H, \ \forall \lambda \in [0, 1],$$

provided that $R > 0$ is sufficiently large. Furthermore, following the same ideas in [1,3–7,16,20], we can conclude that problem (8) has the solution $(U, \zeta) \in W_p^{1,2}(Q) \times W_p^{1,2}(\Sigma)$.

Making use of the embedded $L^{3p-2}(Q) \subset L^p(Q)$ and the estimate (34), it follows that (11) and this completes the proof of the first part in Theorem 1.

*Proof of Theorem 1 Continued*

In this subsection we demonstrate the second part of Theorem 1 which entails checking (14) and thus the uniqueness of the solution to (1) (or (3)). We consider $(U^1, \zeta^1)$ and $(U^2, \zeta^2)$ as in the statement of Theorem 1. From the first part we know that $U^1, U^2 \in W_p^{1,2}(Q)$ and $\zeta^1, \zeta^2 \in W_p^{1,2}(\Sigma)$. Therefore, $U = U^1 - U^2 \in W_p^{1,2}(Q)$ and $Z = \zeta^1 - \zeta^2 \in W_p^{1,2}(\Sigma)$.

Following [1–3,5–7,16,20], the increments of $a_{ij}$ and $A$ (see (4)) can be written in the following form

$$a_{ij}(s, x, U^1, U_x^1) - a_{ij}(s, x, U^2, U_x^2) = \int_0^1 \frac{d}{d\lambda} a_{i,j}\left(s, x, U^\lambda, U_x^\lambda\right) d\lambda,$$

$$A(s, x, U^1, U_x^1) - A(s, x, U^2, U_x^2) = \int_0^1 \frac{d}{d\lambda} A\left(s, x, U^\lambda, U_x^\lambda\right) d\lambda$$

and so

$$a_{ij}(s, x, U^1, U_x^1) U_{x_i x_j}^1 - a_{ij}(s, x, U^2, U_x^2) U_{x_i x_j}^2$$

$$= a_{ij}(s, x, U^1, U_x^1) U_{x_i x_j} + \left\{ U_{x_i x_j}^2 \int_0^1 \frac{\partial}{\partial U_{x_j}^\lambda} a_{i,j}\left(s, x, U^\lambda, U_x^\lambda\right) d\lambda \right\} U_{x_i}, \tag{35}$$

$$A(s, x, U^1, U_x^1) - A(s, x, U^2, U_x^2) = \left\{ \int_0^1 \frac{\partial}{\partial U_{x_j}^\lambda} A\left(s, x, U^\lambda, U_x^\lambda\right) d\lambda \right\} U_{x_i}, \tag{36}$$

where

$$a_{i,j}\left(s, x, U_x^\lambda, U_x^\lambda\right) = \frac{\partial}{\partial U_{x_j}^\lambda}\left[K(s, x, U^\lambda) U_{x_i}^\lambda\right],$$

$$A\left(s, x, U^\lambda, U_x^\lambda\right) = a_i\left(s, x, U^\lambda, U_x^\lambda\right), \quad a_i\left(s, x, U^\lambda, U_x^\lambda\right) = \frac{\partial}{\partial x_i}\left[K(s, x, U^\lambda) U_{x_i}^\lambda\right],$$

$$U^\lambda(s, x) = \lambda U^1(s, x) + (1 - \lambda) U^2(s, x) \quad \text{and}$$

$$U_x^\lambda(s, x) = \lambda U_x^1(s, x) + (1 - \lambda) U_x^2(s, x).$$

Subtracting (3) for $U^2(s, x)$ from (3) for $U^1(s, x)$ and using (35) and (36), we obtain the following linear parabolic problem with inhomogeneous dynamic boundary conditions, i.e.,

$$
\begin{cases}
p_1 \dfrac{\partial}{\partial t} U - \hat{a}_{ij}(s, x) \Delta U = -\hat{a}_i(s, x) \nabla U - p_2 U + p_s(g_d^1 - g_d^2) & \text{in } Q \\[2mm]
U(s, x) = Z(s, x) & \text{on } \Sigma \\[2mm]
U(0, x) = (U_0^1 - U_0^2)(x) & \text{in } \Omega \\[2mm]
p_1 \dfrac{\partial}{\partial \mathbf{n}} U + p_2 \dfrac{\partial}{\partial t} Z - \Delta_\Gamma Z + p_t Z = g_{fr}^1 - g_{fr}^2 & \text{on } \Sigma \\[2mm]
Z(0, x) = (\zeta_0^1 - \zeta_0^2)(x) & \text{on } \Gamma,
\end{cases}
\tag{37}
$$

where

$$\hat{a}_{ij}(s,x) = a_{ij}(s,x,U^1,U^1_x),$$

$$\hat{a}_i(s,x) = -U^2_{x_ix_j}\int_0^1 \frac{\partial}{\partial U^\lambda_{x_j}}a_{i,j}\left(s,x,U^\lambda,U^\lambda_x\right)d\lambda + \int_0^1 \frac{\partial}{\partial U^\lambda_{x_j}}\frac{\partial}{\partial x_i}\left[K(s,x,U^\lambda)U^\lambda_{x_i}\right]d\lambda.$$

Next, following the work of A. Miranville and C. Moroşanu [3], we easily deduce the validity of the estimate in (14); thus, the uniqueness of the solution to (1) or (3) is true. $\square$

**Corollary 1.** *Corresponding to $U^1_0 = U^2_0$ and $\zeta^1_0 = \zeta^2_0$, the problem (1) possesses a unique classical solution.*

## 4. Approximating Scheme—Convergence and Error Estimate

Here we use the fractional steps method in order to approximate the unique solution to problem (8) with inhomogeneous dynamic boundary conditions (see Corollary 1). Precisely, $\forall\, \varepsilon > 0,$ let $M_\varepsilon = \left[\frac{T}{\varepsilon}\right]$ and

$$Q^\varepsilon_i = [i\varepsilon,(i+1)\varepsilon]\times\Omega, \quad \Sigma^\varepsilon_i = [i\varepsilon,(i+1)\varepsilon]\times\partial\Omega \quad i = 0,1,\cdots,M_\varepsilon-1,$$

with $Q^\varepsilon_{M_\varepsilon-1} = [(M_\varepsilon-1)\varepsilon,T]\times\Omega$, $\Sigma^\varepsilon_{M_\varepsilon-1} = [(M_\varepsilon-1)\varepsilon,T]\times\partial\Omega$. Correspondingly, we link the following numerical scheme with problem (8)

$$\begin{cases} p_1\dfrac{\partial}{\partial t}U^\varepsilon - p_2\mathrm{div}\left(K(t,x,U^\varepsilon)\,\nabla U^\varepsilon\right) = p_r U^\varepsilon + p_s g_d(t,x) & \text{in } Q^\varepsilon_i \\[2mm] p_2\dfrac{\partial}{\partial \mathbf{n}}U^\varepsilon + p_1\dfrac{\partial}{\partial t}\zeta^\varepsilon - \Delta_\Gamma\zeta^\varepsilon + p_t\zeta^\varepsilon = g_{fr}(t,x) & \text{on } \Sigma^\varepsilon_i \\[2mm] U^\varepsilon(i\varepsilon,x) = z(\varepsilon,U^\varepsilon_-(i\varepsilon,x)) & \text{on } \Omega \\[2mm] \zeta^\varepsilon(i\varepsilon,x) = U^\varepsilon(i\varepsilon,x) & \text{on } \partial\Omega, \end{cases} \tag{38}$$

with $z(\varepsilon,U^\varepsilon_-(i\varepsilon,x))$ being the solution of Cauchy problem:

$$\begin{cases} z'(s) + p_r z^3(s) = 0 & s\in[0,\varepsilon] \\[2mm] z(0) = U^\varepsilon_-(i\varepsilon,x) & \text{on } \Omega \\[2mm] U^\varepsilon_-(0,x) = U_0(x) & \text{on } \Omega \\[2mm] U^\varepsilon_-(0,x) = \zeta_0(x) & \text{on } \partial\Omega, \end{cases} \tag{39}$$

where $U^\varepsilon_-$ stands for the left-hand limit of $U^\varepsilon$.

For a detailed discussion regarding the importance of the above numerical scheme we direct the reader to the works [5,9,11–14,17–19,22,23].

The main question of this work concerns the convergence as $\varepsilon \to 0$ of the sequence $(U^\varepsilon,\zeta^\varepsilon)$ of the solutions to problems (38) and (39), and to the solution $(U,\zeta)$ of problem (8) (see [11] for more details).

For simplicity, we note:

$$W_Q = L^2([0,T];H^1(\Omega))\cap L^\infty(Q) \quad \text{and} \quad W_\Sigma = L^2([0,T];H^1(\partial\Omega))\cap L^\infty(\Sigma).$$

**Definition 2.** *By a weak solution to problem* (8) *we refer to a pair of functions* $(U, \zeta) \in W_Q \times W_\Sigma$ *and* $U = \zeta$ *on* $\Sigma$, *which satisfy* (8) *in the following sense:*

$$
p_1 \int_Q \left( \frac{\partial}{\partial t} U, \phi_1 \right) dt\, dx + p_2 \int_Q K(t, x, U) \nabla U \cdot \nabla \phi_1 \, dt\, dx
$$

$$
+ p_2 \int_\Sigma \left( \frac{\partial}{\partial t} \zeta, \phi_2 \right) dt\, d\gamma + \int_\Sigma \nabla \zeta \nabla \phi_2 \, dt\, d\gamma + p_t \int_\Sigma \zeta \phi_2 \, dt d\gamma
$$

$$
= p_r \int_Q (U - U^3) \phi_1 \, dt\, dx + p_s \int_Q g_d \phi_1 \, dt\, dx + \int_\Sigma g_{fr} \phi_2 \, dt d\gamma
$$

(40)

$$
\forall (\phi_1, \phi_2) \in L^2([0, T]; H^1(\Omega)) \times L^2([0, T]; H^1(\Gamma)),
$$

*where* $\phi_1 = \phi_2$ *on* $\Sigma$, *and* $U(0, x) = U_0(x)$ *on* $\Omega$.

**Definition 3.** *By a weak solution to problems* (38) *and* (39) *we refer to a pair of functions* $(U^\varepsilon, \zeta^\varepsilon) \in W_{Q_i^\varepsilon} \times W_{\Sigma_i^\varepsilon}$, *and* $U_i^\varepsilon = \zeta_i^\varepsilon$ *on* $\Sigma_i^\varepsilon$, $i \in \{0, 1, \dots, M_\varepsilon - 1\}$, *which satisfy* (38) *and* (39) *in the following sense:*

$$
p_1 \int_Q \left( \frac{\partial}{\partial t} U^\varepsilon, \xi_1 \right) dt\, dx + p_2 \int_Q K(t, x, U^\varepsilon) \nabla U^\varepsilon \cdot \nabla \xi_1 \, dt\, dx
$$

$$
+ p_2 \int_\Sigma \left( \frac{\partial}{\partial t} \zeta^\varepsilon, \xi_2 \right) dt\, d\gamma + \int_\Sigma \nabla \zeta^\varepsilon \nabla \xi_2 \, dt\, d\gamma + p_t \int_\Sigma \zeta^\varepsilon \xi_2 \, dt d\gamma
$$

$$
= p_r \int_Q U^\varepsilon \xi_1 \, dt\, dx + p_s \int_Q g_d \xi_1 \, dt\, dx + \int_\Sigma g_{fr} \xi_2 \, dt d\gamma
$$

(41)

$$
\forall (\xi_1, \xi_2) \in L^2([0, T]; H^1(\Omega)) \times L^2([0, T]; H^1(\partial\Omega)),
$$

*where* $U_-^\varepsilon(0, x) = U_0(x)$ *on* $\Omega$, *and* $U_-^\varepsilon(0, x) = \zeta_0(x)$ *on* $\partial\Omega$.

In (40) and (41) the symbols $\int_Q$ and $\int_\Sigma$ denote the duality between $L^2([0, T]; H^1(\Omega))$ and $L^2([0, T]; H^1(\Omega)')$ as well as $L^2([0, T]; H^1(\partial\Omega))$ and $L^2([0, T]; H^1(\partial\Omega)')$, respectively.

*Convergence of the Numerical Schemes* (38) *and* (39)

The purpose of this subsection is to prove the convergence of the solution to the numerical scheme associated with the non-linear problem (8). Therefore,

**Theorem 2.** *Assume that* $U_0(x) \in W_\infty^{2 - \frac{2}{2}}(\Omega)$, *satisfying* $p_2 \frac{\partial}{\partial \nu} U_0 - \Delta_\Gamma U_0 + p_t U_0 = g_{fr}(0, x)$ *on* $\partial\Omega$ *and* $g_{fr}(s, x) \in W_p^{1 - \frac{1}{2p}, 2 - \frac{1}{p}}(\Sigma)$. *Let* $(U^\varepsilon, \zeta^\varepsilon)$ *be the solution to the numerical schemes* (38) *and* (39). *As* $\varepsilon \to 0$, *one has*

$$
(U^\varepsilon, \zeta^\varepsilon) \to (U^\star, \zeta^\star) \quad \text{strongly in } L^2(\Omega) \times L^2(\partial\Omega) \text{ for any } s \in (0, T],
$$

(42)

*where* $(U^\star, \zeta^\star) \in L^2([0, T]; H^1(\Omega)) \times L^2([0, T]; H^1(\partial\Omega))$ *is a weak solution to problem* (8).

The following lemmas, which involve the Cauchy problem (39), are very useful in the proof of Theorem 2. These were proven for the first time in [11]. Here, we reproduce them as well as sketch out the proof when pertinent.

**Lemma 1.** *Assume $U^\varepsilon_-(i\varepsilon, x) \in L^\infty(\Omega)$, $i = 0, 1, \ldots, M_\varepsilon - 1$. Then, $U^\varepsilon(i\varepsilon, x) \in L^\infty(\Omega)$ and*

$$\|U^\varepsilon(i\varepsilon, x)\|^2_{L^2(\Omega)} \leq \|U^\varepsilon_-(i\varepsilon, x)\|^2_{L^2(\Omega)}. \tag{43}$$

**Proof.** We write $(39)_1$ in the form $\left(\dfrac{1}{z^2}\right)' = p_r$, and following the same reasoning as in [11] we obtain

$$z^2(\varepsilon, U^\varepsilon_-(i\varepsilon, x)) \leq U^\varepsilon_-(i\varepsilon, x)^2, \quad a.e \ x \in \Omega. \tag{44}$$

Owing to $(38)_3$ and (44), we can easily conclude the inequality complete in (43). $\square$

**Lemma 2.** *For $i = 0, 1, \ldots, M_\varepsilon - 1$, the estimate below holds*

$$\|\nabla U^\varepsilon(i\varepsilon, x)\|_{L^2(\Omega)} \leq \|\nabla U^\varepsilon_-(i\varepsilon, x)\|_{L^2(\Omega)}. \tag{45}$$

**Lemma 3.** *The following estimate holds*

$$\|z(\varepsilon, x) - U^\varepsilon_-(i\varepsilon, x)\|_{L^2(\Omega)} \leq \varepsilon L \tag{46}$$

*where $L > 0$ depends on $|\Omega|$, $\|U^\varepsilon_-\|_{L^\infty(\Omega)}$ and $p_2$.*

Now, we are in a position to give the proof of Theorem 2. Following the same steps as in [11], we obtain the solution to problem (38) as $(U^\varepsilon, \zeta^\varepsilon) \in W^{1,2}_p(Q^\varepsilon_i) \cap L^\infty(Q^\varepsilon_i) \times W^{1,2}_p(\Sigma^\varepsilon_i) \cap L^\infty(\Sigma^\varepsilon_i)$, $\forall i \in \{0, 1, \ldots, M_\varepsilon - 1\}$.

Next, we give a priori estimates to $Q^\varepsilon_i$, $\forall i \in \{0, 1, \ldots, M_\varepsilon - 1\}$. Firstly, we multiply $(38)_1$ by $U^\varepsilon_t$ and obtain

$$p_1 \int_\Omega |U^\varepsilon_t|^2 dx + p_1 \int_\Gamma |\zeta^\varepsilon_t|^2 d\gamma$$

$$+ \frac{p_2}{2} \int_\Omega K(t, x, U^\varepsilon) \frac{d}{dt} |\nabla U^\varepsilon|^2 dx + \frac{1}{2} \frac{d}{dt} \int_\Gamma |\nabla_\Gamma \zeta^\varepsilon|^2 d\gamma + \frac{p_t}{2} \frac{d}{dt} \int_\Gamma |\zeta^\varepsilon|^2 d\gamma \tag{47}$$

$$= \frac{p_2}{2} \frac{d}{dt} \int_\Omega |U^\varepsilon|^2 dx + \int_\Gamma g_{fr} \zeta^\varepsilon_t d\gamma + p_s \int_\Omega g_d U^\varepsilon_t dx.$$

Using Hölder's inequality for the right-hand terms $\int_\Gamma g_{fr} \zeta^\varepsilon_t d\gamma$ and $\int_\Omega g_d U^\varepsilon_t dx$, we have

$$\int_\Gamma g_{fr} \zeta^\varepsilon_t d\gamma \leq \frac{p_1}{2} \int_\Gamma |\zeta^\varepsilon_t|^2 d\gamma + \frac{1}{2p_1} \int_\Gamma |g_{fr}|^2 d\gamma,$$

$$p_s \int_\Omega g_d U^\varepsilon_t dx \leq \frac{p_1}{2} \int_\Omega |U^\varepsilon_t|^2 dx + \frac{p_s}{2p_1} \int_\Omega |g_d|^2 dx,$$

and substituting them in (47), we derive

$$
\frac{p_1}{2} \int_\Omega |U_t^\varepsilon|^2 dx + \frac{p_1}{2} \int_\Gamma |\zeta_t^\varepsilon|^2 d\gamma
$$

$$
+ \frac{p_2}{2} K_{min} \frac{d}{dt} \int_\Omega |\nabla U^\varepsilon|^2 dx + \frac{1}{2}\frac{d}{dt}\int_\Gamma |\nabla_\Gamma \zeta^\varepsilon|^2 d\gamma + \frac{p_t}{2}\frac{d}{dt}\int_\Gamma |\zeta^\varepsilon|^2 d\gamma \tag{48}
$$

$$
\leq \frac{p_2}{2}\frac{d}{dt}\int_\Omega |U^\varepsilon|^2 dx + \frac{1}{2p_1}\int_\Gamma |g_{fr}|^2\, d\gamma + \frac{p_s}{2p_1}\int_\Omega |g_d|^2\, dx,
$$

where the inequality (9) is also used.

Multiplying $(38)_1$ by $\frac{1}{p_1 p_2} U^\varepsilon$ as shown above, we obtain

$$
\frac{1}{2p_2}\frac{d}{dt}\int_\Omega |U^\varepsilon|^2 dx + \frac{1}{2p_2}\frac{d}{dt}\int_\Gamma |\zeta^\varepsilon|^2 d\gamma
$$

$$
+ \frac{1}{p_1}\int_\Omega K(t,x,U^\varepsilon)|\nabla U^\varepsilon|^2 dx + \frac{1}{p_1}\int_\Gamma |\nabla_\Gamma \zeta^\varepsilon|^2 d\gamma + \frac{p_t}{p_1 p_2}\int_\Gamma |\zeta^\varepsilon|^2 d\gamma \tag{49}
$$

$$
= \frac{1}{p_1 p_2 p_r}\int_\Omega |U^\varepsilon|^2 dx + \frac{1}{p_1 p_2}\int_\Gamma g_{fr}\zeta^\varepsilon\, d\gamma + \frac{p_s}{p_1 p_2}\int_\Omega g_d U^\varepsilon\, dx.
$$

In addition, using Hölder's inequality for the right-hand terms $\int_\Gamma g_{fr}\zeta^\varepsilon\, d\gamma$ and $\int_\Omega g_d U^\varepsilon\, dx$, we have

$$
\frac{1}{p_1 p_2}\int_\Gamma g_{fr}\zeta^\varepsilon\, d\gamma \leq \frac{2p_t}{p_1 p_2}\int_\Gamma |\zeta^\varepsilon|^2\, d\gamma + \frac{1}{2p_t p_1 p_2}\int_\Gamma |g_{fr}|^2\, d\gamma,
$$

$$
\frac{p_s}{p_1 p_2}\int_\Omega g_d U^\varepsilon\, dx \leq \frac{1}{p_1 p_2}\int_\Omega |U^\varepsilon|^2\, dx + \frac{p_s}{p_1 p_2}\int_\Omega |g_d|^2\, dx,
$$

and then from (49) we obtain

$$
\frac{1}{2p_2}\frac{d}{dt}\int_\Omega |U^\varepsilon|^2 dx + \frac{1}{2p_2}\frac{d}{dt}\int_\Gamma |\zeta^\varepsilon|^2 d\gamma
$$

$$
+ \frac{1}{p_1} K_{min} \int_\Omega |\nabla U^\varepsilon|^2 dx + \frac{1}{p_1}\int_\Gamma |\nabla_\Gamma \zeta^\varepsilon|^2 d\gamma \tag{50}
$$

$$
\leq C(p_s, p_t, p_1, p_2)\left[\int_\Omega |U^\varepsilon|^2 dx + \int_\Gamma |\zeta^\varepsilon|^2\, d\gamma + \int_\Gamma |g_{fr}|^2\, d\gamma + \int_\Omega |g_d|^2\, dx\right],
$$

where the inequality (9) is also used.

Adding (48) and (50), we obtain

$$\frac{\partial}{\partial t}\left[\frac{1}{2p_2}\int_\Omega |U^\varepsilon|^2 dx + \left(\frac{p_t}{2}+\frac{1}{2p_2}\right)\int_\Gamma |\zeta^\varepsilon|^2 d\gamma + \frac{p_2}{2}K_{min}\int_\Omega |\nabla U^\varepsilon|^2 dx + \frac{1}{2}\int_\Gamma |\nabla_\Gamma \zeta^\varepsilon|^2 dx\right]$$

$$+ \frac{p_1}{2}\int_\Omega |U_t^\varepsilon|^2 dx + \frac{p_1}{2}\int_\Gamma |\zeta_t^\varepsilon|^2 d\gamma + \frac{K_{min}}{p_1}\int_\Omega |\nabla U^\varepsilon|^2 dx + \frac{1}{p_1}\int_\Gamma |\nabla_\Gamma \zeta^\varepsilon|^2 d\gamma$$

$$\leq C(p_s, p_t, p_1, p_2)\left[\int_\Omega |U^\varepsilon|^2 dx + \int_\Gamma |\zeta^\varepsilon|^2 d\gamma + \int_\Gamma |g_{f_r}|^2 d\gamma + \int_\Omega |g_d|^2 dx\right].$$

Integrating the preceding on $Q_0^\varepsilon$, we derive

$$\frac{1}{2p_2}\|U_-^\varepsilon(\varepsilon, x)\|_{L^2(\Omega)}^2 + \left(\frac{p_t}{2}+\frac{1}{2p_2}\right)\|\zeta_-^\varepsilon(\varepsilon, x)\|_{L^2(\Gamma)}^2$$

$$+ \frac{p_2}{2}K_{min}\|\nabla U_-^\varepsilon(\varepsilon, x)\|_{L^2(\Omega)}^2 + \frac{1}{2}\|\nabla_\Gamma \zeta_-^\varepsilon(\varepsilon, x)\|_{L^2(\Gamma)}^2$$

$$+ \int_0^\varepsilon\left[\frac{p_1}{2}\int_\Omega |U_t^\varepsilon|^2 dx + \frac{p_1}{2}\int_\Gamma |\zeta_t^\varepsilon|^2 d\gamma + \frac{K_{min}}{p_1}\int_\Omega |\nabla U^\varepsilon|^2 dx + \frac{1}{p_1}\int_\Gamma |\nabla_\Gamma \zeta^\varepsilon|^2 d\gamma\right]ds \qquad (51)$$

$$\leq \frac{1}{2p_2}\|U_0\|_{L^2(\Omega)}^2 + \left(\frac{p_t}{2}+\frac{1}{2p_2}\right)\|\zeta_0\|_{L^2(\Gamma)}^2 + \frac{p_2}{2}K_{min}\|\nabla U_0\|_{L^2(\Omega)}^2 + \frac{1}{2}\|\nabla_\Gamma \zeta_0\|_{L^2(\Gamma)}^2$$

$$+ C(p_s, p_t, p_1, p_2)\left\{\int_0^\varepsilon\left[\|U^\varepsilon\|_{L^2(\Omega)}^2 + \|\zeta^\varepsilon\|_{L^2(\Gamma)}^2\right]ds + \|g_{f_r}\|_{L^2(\Sigma_0^\varepsilon)}^2 + \|g_d\|_{L^2(Q_0^\varepsilon)}^2\right\}.$$

It is relatively easy to observe that the estimate above refers to $Q_0^\varepsilon$ and $\Sigma_0^\varepsilon$ ($i = 0$). Proceeding in a similar way for $i = 1, 2, \ldots, M_\varepsilon - 2$, we obtain

$$\frac{1}{2p_2}\|U_-^\varepsilon((i+1)\varepsilon, x)\|_{L^2(\Omega)}^2 + \left(\frac{p_t}{2}+\frac{1}{2p_2}\right)\|\zeta_-^\varepsilon((i+1)\varepsilon, x)\|_{L^2(\Gamma)}^2$$

$$+ \frac{p_2}{2}K_{min}\|\nabla U_-^\varepsilon((i+1)\varepsilon, x)\|_{L^2(\Omega)}^2 + \frac{1}{2}\|\nabla_\Gamma \zeta_-^\varepsilon((i+1)\varepsilon, x)\|_{L^2(\Gamma)}^2$$

$$+ \int_{i\varepsilon}^{(i+1)\varepsilon}\left[\frac{p_1}{2}\|U_t^\varepsilon\|_{L^2(\Omega)}^2 + \frac{p_1}{2}\|\zeta_t^\varepsilon\|_{L^2(\Gamma)}^2 + \frac{K_{min}}{p_1}\|\nabla U^\varepsilon\|_{L^2(\Omega)}^2 + \frac{1}{p_1}\|\nabla_\Gamma \zeta^\varepsilon\|_{L^2(\Gamma)}^2\right]ds \qquad (52)$$

$$\leq \frac{1}{2p_2}\|U^\varepsilon(i\varepsilon, x)\|_{L^2(\Omega)}^2 + \left(\frac{p_t}{2}+\frac{1}{2p_2}\right)\|\zeta^\varepsilon(i\varepsilon, x)\|_{L^2(\Gamma)}^2$$

$$+ \frac{p_2}{2}\|\nabla U^\varepsilon(i\varepsilon, x)\|_{L^2(\Omega)}^2 + \frac{1}{2}\|\nabla_\Gamma \zeta^\varepsilon(i\varepsilon, x)\|_{L^2(\Gamma)}^2$$

$$+ C(p_s, p_t, p_1, p_2)\left\{\int_{i\varepsilon}^{(i+1)\varepsilon}\left[\|U^\varepsilon\|_{L^2(\Omega)}^2 + \|\zeta^\varepsilon\|_{L^2(\Gamma)}^2\right]ds + \|g_{f_r}\|_{L^2(\Sigma_i^\varepsilon)}^2 + \|g_d\|_{L^2(Q_i^\varepsilon)}^2\right\},$$

while for $i = M_\varepsilon - 1$ we have

$$
\frac{1}{2p_2}\|U_-^\varepsilon(T,x)\|_{L^2(\Omega)}^2 + \left(\frac{p_t}{2} + \frac{1}{2p_2}\right)\|\zeta_-^\varepsilon(T,x)\|_{L^2(\Gamma)}^2
$$

$$
+ \frac{p_2}{2}K_{min}\|\nabla U_-^\varepsilon(T,x)\|_{L^2(\Omega)}^2 + \frac{1}{2}\|\nabla_\Gamma\zeta_-^\varepsilon(T,x)\|_{L^2(\Gamma)}^2
$$

$$
+ \int_{M_\varepsilon-1}^{T}\left[\frac{p_1}{2}\|U_t^\varepsilon\|_{L^2(\Omega)}^2 + \frac{p_1}{2}\|\zeta_t^\varepsilon\|_{L^2(\Gamma)}^2 + \frac{1}{p_1}\|\nabla U^\varepsilon\|_{L^2(\Omega)}^2 + \frac{1}{p_1}\|\nabla_\Gamma\zeta^\varepsilon\|_{L^2(\Gamma)}^2\right]ds
$$

$$
\leq \frac{1}{2p_2}\|U^\varepsilon(T,x)\|_{L^2(\Omega)}^2 + \left(\frac{p_t}{2} + \frac{1}{2p_2}\right)\|\zeta^\varepsilon(T,x)\|_{L^2(\Gamma)}^2
$$

$$
+ \frac{p_2}{2}\|\nabla U^\varepsilon(T,x)\|_{L^2(\Omega)}^2 + \frac{1}{2}\|\nabla_\Gamma\zeta^\varepsilon(T,x)\|_{L^2(\Gamma)}^2
$$

$$
+ C(p_s,p_t,p_1,p_2)\left\{\int_{M_\varepsilon-1}^{T}\left[\|U^\varepsilon\|_{L^2(\Omega)}^2 + \|\zeta^\varepsilon\|_{L^2(\Gamma)}^2\right]ds + \|g_{fr}\|_{L^2(\Sigma_{M_\varepsilon-1}^\varepsilon)}^2 + \|g_d\|_{L^2(Q_{M_\varepsilon-1}^\varepsilon)}^2\right\}. \tag{53}
$$

Adding (51)–(53) and owing to the inequalities (43) and (45), we obtain

$$
\frac{1}{2p_2}\|U_-^\varepsilon(T,x)\|_{L^2(\Omega)}^2 + \left(\frac{p_t}{2} + \frac{1}{2p_2}\right)\|\zeta_-^\varepsilon(T,x)\|_{L^2(\Gamma)}^2
$$

$$
+ \frac{p_2}{2}\|\nabla U_-^\varepsilon(T,x)\|_{L^2(\Omega)}^2 + \frac{1}{2}\|\nabla_\Gamma\zeta_-^\varepsilon(T,x)\|_{L^2(\Gamma)}^2
$$

$$
+ \int_0^{T}\left[\frac{p_1}{2}\|U_t^\varepsilon\|_{L^2(\Omega)}^2 + \frac{p_1}{2}\|\zeta_t^\varepsilon\|_{L^2(\Gamma)}^2 + \frac{1}{p_1}\|\nabla U^\varepsilon\|_{L^2(\Omega)}^2 + \frac{1}{p_1}\|\nabla_\Gamma\zeta^\varepsilon\|_{L^2(\Gamma)}^2\right]dt
$$

$$
\leq \frac{1}{2p_2}\|U_0\|_{L^2(\Omega)}^2 + \left(\frac{p_t}{2} + \frac{1}{2p_2}\right)\|\psi_0\|_{L^2(\Gamma)}^2 + \frac{p_2}{2}\|\nabla U_0\|_{L^2(\Omega)}^2 + \frac{1}{2}\|\nabla_\Gamma\zeta_0\|_{L^2(\Gamma)}^2
$$

$$
+ C(p_s,p_t,p_1,p_2)\left\{\int_0^{T}\left[\|U^\varepsilon\|_{L^2(\Omega)}^2 + \|\zeta^\varepsilon\|_{L^2(\Gamma)}^2\right]dt + \|g_{fr}\|_{L^2(\Sigma)}^2 + \|g_d\|_{L^2(Q)}^2\right\}.
$$

Applying the Gronwall inequality to the above inequalities, we finally deduce

$$
\int_0^{T}\left\{\|U_t^\varepsilon\|_{L^2(\Omega)}^2 + \|\zeta_t^\varepsilon\|_{L^2(\Gamma)}^2 + \|\nabla U^\varepsilon\|_{L^2(\Omega)}^2 + \|\nabla_\Gamma\zeta^\varepsilon\|_{L^2(\Gamma)}^2\right\}dt \leq C, \tag{54}
$$

where $C > 0$ is independent of $\varepsilon$ and $M_\varepsilon$.

Owing to (38)$_3$, (38)$_4$ and (46), we obtain

$$
\sum_{i=0}^{M_\varepsilon-1}\|U^\varepsilon(i\varepsilon,x) - U_-^\varepsilon(i\varepsilon,x)\|_{L^2(\Omega)} \leq TL = C_1, \tag{55}
$$

$$\sum_{i=0}^{M_\varepsilon - 1} \|\zeta^\varepsilon(i\varepsilon, x) - \zeta_-^\varepsilon(i\varepsilon, x)\|_{L^2(\Gamma)} \leq C_2, \tag{56}$$

where $C_1 > 0$ and $C_2 > 0$ are independent of $M_\varepsilon$ and $\varepsilon$. Summing (54)–(56), we derive

$$\overset{T}{\underset{0}{V1}}\, U^\varepsilon + \overset{T}{\underset{0}{V2}}\, \zeta^\varepsilon + \int_0^T \left[\|U_t^\varepsilon\|_{L^2(\Omega)}^2 + \|\zeta_t^\varepsilon\|_{L^2(\Gamma)}^2 + \|\nabla U^\varepsilon\|_{L^2(\Omega)}^2 + \|\nabla_\Gamma \zeta^\varepsilon\|_{L^2(\Gamma)}^2\right] ds \leq C, \tag{57}$$

where the positive constant $C$ is independent of $M_\varepsilon$ and $\varepsilon$, while $\overset{T}{\underset{0}{V1}}\, U^\varepsilon$ and $\overset{T}{\underset{0}{V2}}\, \zeta^\varepsilon$ stand for the variation of $U^\varepsilon : [0, T] \to L^2(\Omega)$ and $\zeta^\varepsilon : [0, T] \to L^2(\Gamma)$, respectively.

Since the introduction of $L^2(\Omega)$ into $H^{-1}(\Omega)$ is compact and $\{U_s^\varepsilon(s)\}$ is bounded in $L^2(\Omega)\ \forall s \in [0, T]$, we conclude that there exists a bounded variation function $U^*(s) \in BV([0, T]; H^{-1}(\Omega))$ and subsequent $U^\varepsilon(s)$ (see [11]), such that

$$U^\varepsilon(s) \to U^*(s) \quad \text{strongly in} \quad H^{-1}(\Omega) \quad \forall s \in [0, T], \tag{58}$$

$$\zeta^\varepsilon(s) \to \zeta^*(s) \quad \text{strongly in} \quad H^{-1}(\Gamma) \quad \forall s \in [0, T]. \tag{59}$$

Further, from (57) we deduce that

$$\begin{cases} U^\varepsilon \to U^* & \text{weakly in } L^2(0, T; H^1(\Omega)) \\ \zeta^\varepsilon \to \zeta^* & \text{weakly in } L^2(0, T; H^1(\Gamma)). \end{cases} \tag{60}$$

By the well-known embeddings $H^1(\Omega) \subset L^2(\Omega) \subset H^{-1}(\Omega)$, and $H^1(\partial\Omega) \subset L^2(\partial\Omega) \subset H^{-1}(\partial\Omega)$, standard interpolation inequalities (see [11] p. 17) yield that $\forall \ell > 0, \exists C(\ell) > 0$ such that

$$\begin{cases} \|U^\varepsilon(s) - U^*(s)\|_{L^2(\Omega)} \leq \ell \|U^\varepsilon(s) - U^*(s)\|_{H^1(\Omega)} + C(\ell)\|U^\varepsilon(s) - U^*(s)\|_{H^{-1}(\Omega)}, \\ \|\zeta^\varepsilon(s) - \zeta^*(s)\|_{L^2(\partial\Omega)} \leq \ell \|\zeta^\varepsilon(s) - \zeta^*(s)\|_{H^1(\partial\Omega)} + C(\ell)\|\zeta^\varepsilon(s) - \zeta^*(s)\|_{H^{-1}(\partial\Omega)}, \end{cases} \tag{61}$$

$\forall \varepsilon > 0$ and $\forall s \in [0, T]$, where $C(\ell) \to 0$ as $\ell \to 0$.

Finally, relations (58)–(61) permit us to conclude that the assertion conducted in (42) holds true, ending the proof of Theorem 2.

**Corollary 2.** *Assume $U_0 \in W_\infty^{2 - \frac{2}{p}}(\Omega)$, $p_2 \frac{\partial}{\partial \nu} U_0(x) - \Delta_\Gamma U_0 + p_t U_0(x) = g_{fr}(0, x)$ on $\partial\Omega$ and $g_{fr} \in W_p^{1 - \frac{1}{2p}, 2 - \frac{1}{p}}(\Sigma)$. Then $U^\star \in W_Q$ is a weak solution to the non-linear problem in (1).*

Now we search the error of the numerical schemes (38) and (39) relative to $g_d$ and $g_{fr}$. From Theorem 1 we know that $\forall g_d \in L^p(Q)$ and $g_{fr} \in W_p^{1 - \frac{1}{2p}, 2 - \frac{1}{p}}(\Sigma)$, the problem (8) has a unique solution $(U, \zeta) \in W_p^{1,2}(Q) \times W_p^{1,2}(\Sigma)$. Moreover, (see (11))

$$\|U\|_{W_p^{1,2}(Q)} + \|\zeta\|_{W_p^{1,2}(\Sigma)}$$

$$\leq C\left[1 + \|U_0\|_{W_\infty^{2 - \frac{2}{p}}(\Omega)}^{3 - \frac{2}{p}} + \|\zeta_0\|_{W_\infty^{2 - \frac{2}{p}}(\Gamma)}^{3 - \frac{2}{p}} + \|g_d\|_{L^{3p-2}(Q)}^{\frac{3p-2}{p}} + \|g_{fr}\|_{W_p^{1 - \frac{1}{2p}, 2 - \frac{1}{p}}(\Sigma)}\right], \tag{62}$$

with a fixed $\zeta_0 \in W_\infty^{2 - \frac{2}{p}}(\Gamma)$ and $U_0 \in W_\infty^{2 - \frac{2}{p}}(\Omega)$ verifying $p_2 \frac{\partial}{\partial \nu} U_0 - \Delta_\Gamma U_0 + p_t U_0 = g_{fr}(0, x)$. Thus, we have

**Theorem 3.** *Let $g_d \in L^p(Q)$ and $g_{fr} \in W_p^{1-\frac{1}{2p},2-\frac{1}{p}}(\Sigma)$. Let $g_d^k \subset L^p(Q)$ and $g_{fr}^k \subset W_p^{1-\frac{1}{2p},2-\frac{1}{p}}(\Sigma)$ be two sequences such that $g_d^k \longrightarrow g_d$ in $L^p(Q)$ and $g_{fr}^k \longrightarrow g_{fr}$ in $W_p^{1-\frac{1}{2p},2-\frac{1}{p}}(\Sigma)$ as $k \longrightarrow \infty$. Denoted by $(U_m, \zeta_m) \subset W_p^{1,2}(Q) \times W_p^{1,2}(\Sigma)$ and $(U_{m,k}, \zeta_{m,k}) \subset W_p^{1,2}(Q) \times W_p^{1,2}(\Sigma)$, the approximating sequences are given in (38) and ((39), for $(g_d, g_{fr})$ and $(g_d^k, g_{fr}^k)$, respectively, with $U_0 \in W_\infty^{2-\frac{2}{p}}(\Omega)$ fixed. Then,*

$$\limsup_{m \longrightarrow \infty} \left[ \|U_{m,k} - U\|_{L^2(Q)} + \|\zeta_{m,k} - \zeta\|_{L^2(\Sigma)} \right]$$

$$\le Ce^{CT} max \left\{ \max_{(t,x) \in Q} |g_d^k - g_d|, \max_{(t,x) \in \Sigma} |g_{fr}^k - g_{fr}| \right\}$$

(63)

*$\forall k \ge 1$, where $C > 0$ depends on $|\Omega|$, $T$, $n$, $p$, $p_1$, $p_2$, $p_t$, $p_r$, $p_s$, $\|U_0\|_{W_\infty^{2-\frac{2}{p}}(\Omega)}$, $\|g_d\|_{L^p(Q)}$ and $\|g_{fr}\|_{W_p^{1-\frac{1}{2p},2-\frac{1}{p}}(\Sigma)}$.*

In particular, $\exists (U_{m,k}, \zeta_{m,k})$, denoted by $(U_{m_k}, \zeta_{m_k})$, such that $(U_{m_k}, \zeta_{m_k}) \longrightarrow (U, \zeta)$ in $L^p(Q) \times L^p(\Sigma)$ and in $Q \times \Sigma$ as $k \longrightarrow \infty$.

**Proof.** Owing to (62) we assume that

$$\|U_k\|_{W_p^{1,2}(Q)} + \|\zeta_k\|_{W_p^{1,2}(\Sigma)}$$

$$\le C \left\{ 1 + \|U_0\|_{W_\infty^{2-\frac{2}{p}}(\Omega)}^{3-\frac{2}{p}} + \|\zeta_0\|_{W_\infty^{2-\frac{2}{p}}(\Gamma)}^{3-\frac{2}{p}} + \|g_d^k\|_{L^{3p-2}(Q)}^{\frac{3p-2}{p}} + \|g_{fr}^k\|_{W_p^{1-\frac{1}{2p},2-\frac{1}{p}}(\Sigma)} \right\}$$

$$\le C \left\{ 1 + \|U_0\|_{W_\infty^{2-\frac{2}{p}}(\Omega)}^{3-\frac{2}{p}} + \|\zeta_0\|_{W_\infty^{2-\frac{2}{p}}(\Gamma)}^{3-\frac{2}{p}} + \|g_d\|_{L^{3p-2}(Q)}^{\frac{3p-2}{p}} + \|g_{fr}\|_{W_p^{1-\frac{1}{2p},2-\frac{1}{p}}(\Sigma)} \right\},$$

where $C > 0$ is interpreted as $M_4$ in (12). This ensures the applicability of (14) in Theorem 1 with $U_0^1 = U_0^2$ and $\zeta_0^1 = \zeta_0^2$ obtains

$$\|U_k - U\|_{W_p^{1,2}(Q)} + \|\zeta_k - \zeta\|_{W_p^{1,2}(\Sigma)}$$

$$\le C_1 e^{CT} max \left\{ \max_{(t,x) \in Q} |g_d^k - g_d|, \max_{(t,x) \in \Sigma} |g_{fr}^k - g_{fr}| \right\}, \quad \forall k \ge 1,$$

(64)

where $C_1 > 0$. For $k \ge 1$, Theorem 2 gives

$$(U_{m,k}(s, \cdot), \zeta_{m,k}(s, \cdot) \longrightarrow (U_k(s, \cdot), \zeta_k(s, \cdot)) \quad in \quad L^2(\Omega) \times L^2(\partial\Omega),$$

uniformly for $s \in [0, T]$, as $m \longrightarrow \infty$. In particular, $\forall k \ge 1$ we have

$$(U_{m,k}, \zeta_{m,k}) \longrightarrow (U_k, \zeta_k), \quad in \quad L^2(Q) \times L^2(\Sigma), \quad as \quad m \longrightarrow \infty.$$

(65)

On the base of the relation in (64) and owing to (20), we obtain

$$\|U_{m,k} - U\|_{L^2(Q)} + \|\zeta_{m,k} - \zeta\|_{L^2(\Sigma)}$$

$$\leq \|U_{m,k} - U_k\|_{L^2(Q)} + \|\zeta_{m,k} - \zeta_k\|_{L^2(\Sigma)} + \|U_k - U\|_{L^2(Q)} + \|\zeta_k - \zeta\|_{L^2(\Sigma)}$$

$$\leq \|U_{m,k} - U_k\|_{L^2(Q)} + \|\zeta_{m,k} - \zeta_k\|_{L^2(\Sigma)}$$

$$+ C_1 e^{CT} max\left\{ \max_{(t,x)\in Q} |g_d^k - g_d|, \max_{(t,x)\in \Sigma} |g_{fr}^k - g_{fr}| \right\}, \quad \forall m, k \geq 1.$$

Using (65) we can substitute the above inequality into the superior limit as $m \longrightarrow \infty$ to prove that (63) is correct.

The last statement in Theorem 3 follows directly on from (63). □

The general frameworl of the numerical algorithm to compute the approximate solution to problem (1) via the fractional steps scheme may be demonstrated as follows:

> Begin **alg-frac_sec-ord_dbc**
> $i = 0 \rightarrow U_0$ from (39)$_3$;
> For $i = 0$ perform $M_\varepsilon - 1$
> Compute $z(\varepsilon, \cdot)$ from (39);
> $U^\varepsilon(i\varepsilon, \cdot) = z(\varepsilon, \cdot)$;
> $\zeta^\varepsilon(i\varepsilon, \cdot) = U^\varepsilon(i\varepsilon, \cdot)$;
> Compute $(U^\varepsilon((i+1)\varepsilon, \cdot), \zeta^\varepsilon((i+1)\varepsilon, \cdot))$ solving the linear system (38);
> End-for;
> End.

## 5. Conclusions

The main problem addressed in this work concerns the non-linear second-order reaction–diffusion equation with its principal part in divergence form with inhomogeneous dynamic boundary conditions. Provided that the initial and boundary data meet the appropriate regularity and compatibility conditions, the well-posedness of a classical solution to the non-linear problem is proven in this new formulation (Theorem 1). Precisely, the Leray–Schauder principle and $L^p$ theory of linear and quasi-linear parabolic equations, via Lemma 7.4 (see [1]), were applied to prove the qualitative properties of solution $(U(t,x), \zeta(t,x))$. More precisely, we cannot directly apply the $L^p$ theory to problem (1) (or (3)). Thus, this makes the result of Lemma 7.4 in Choban and Moroşanu [1] (p. 114) very important. Moreover, the a priori estimates were made in $L^p(Q)$ and $L^p(\Sigma)$ which permit the derivation of higher-order regularity properties, that is, $\left(U(t,x), \zeta(t,x)\right) \in W_p^{1,2}(Q) \times W_p^{1,2}(\Sigma)$. Thus, the classical method of bootstrapping (see Moroşanu and Motreanu [20]) can be avoided.

Let us note that, due to the presence of the terms $K(t, x, U(t, x))$, the non-linear operator $H$ (see (17)) does not represent the gradient of the energy functional. Therefore, the new proposed second-order non-linear problem cannot be obtained from the minimisation of any energy cost functional, i.e., (1) is not a variational PDE model.

Furthermore, an iterative fractional step-type scheme was introduced to approximate problem (8). The convergence and error estimates were established for the proposed numerical scheme and a conceptual numerical algorithm was formulated. In this regards, we want to underline the solutions dependence in Theorem 2 on the physical parameters, which could be useful in future investigations regarding error analysis and numerical simulations.

The qualitative results obtained here could be later used in quantitative approaches to the mathematical model (1) (or (3)) as well as in the study of distributed and/or non-linear optimal boundary control problems governed by such a non-linear problem.

Numerical implementation of the conceptual algorithm, **alg-frac_sec-ord_dbc**, as well as various simulations regarding the physical phenomena described by the non-linear parabolic problem (1) represent a matter for further investigation.

**Author Contributions:** Conceptualization, C.F. and C.M.; methodology, C.M.; validation, C.F. and C.M.; writing—original draft preparation, C.M.; writing—review and editing, C.F.; visualization, C.F.; funding acquisition, C.F. All authors have read and agreed to the published version of the manuscript.

**Funding:** This research received no external funding.

**Data Availability Statement:** Not applicable.

**Conflicts of Interest:** The authors declare no conflict of interest.

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
