# Peer review of "Fractional Step Scheme to Approximate a Non-Linear Second-Order Reaction–Diffusion Problem with Inhomogeneous Dynamic Boundary Conditions"

_axioms, doi:10.3390/axioms12040406_

Round 1
Reviewer 1 Report
1.English language are minor spell check required.
2.Consider adding more references regarding research in this field.

Author Response
Our answer to Reviewer 1:
1. Some minor typos was fixed, as reviewer indicated, and marked in the work with orange color;
2. The reference:
"Ovono, A. A., Numerical approximation of the phase-field transition system with non-homogeneous Cauchy-Neumann boundary conditions in both unknown functions via fractional steps methods. Journal of Applied Analysis and Computation} 2013, Volume 3(4), 377-397, was added.
Tank you very much for reviewing our paper.
Best regards,
Constantin FETECAU and Costică MOROSANU
Reviewer 2 Report
1. The authors should revise the whole paper.
2. What is the main advantage and disadvantage of the fractional step method?
3. Can the author use another method for the error estimation?
Author Response
Our answer to Reviewer 2:
1. The whole text of the paper was carefully checked; the typos was fixed and marked with orange color;
2. The fractional steps method simplify the numerical computation; no disadvantage we know;
3. Not for the moment; may be a challenge for future investigations.
Tank you very much for reviewing our paper.
Best regards,
Constantin FETECAU and Costică MOROSANU
Reviewer 3 Report
Fractional steps scheme to approximate a nonlinear second-order reaction-diffusion problem endowed with in-homogeneous dynamic boundary conditions
Authors have studied a nonlinear second-order reaction-diffusion problem in the presented effort. They have performed a rigorous qualitative study for a second-order reaction-diffusion problem along with developing an iterative scheme of fractional steps type. The work is interesting and suitable for publication after below mentioned minor corrections:
· Cite the paper from where the nonlinear second-order reaction-diffusion problem (1) has been taken.
· Mention explicitly that which type of boundary conditions are used to solve the problem.
· There are some typo mistakes which should be corrected.
Author Response
Our answer to Reviewer 3:
1.The formulation of problem (1) is new and, according to our knowledge, has not been treated so far in the mathematics literature;
2. "in-homogeneous dynamic boundary conditions";
3. Some typos was fixed and marked with orange color.
Tank you very much for reviewing our paper.
Best regards,
Constantin FETECAU and Costică MOROSANU
Reviewer 4 Report
Review Report
of
axioms-2276542
In the reviewed work, the authors study nonlinear second-order boundary value problem. Results about existence, uniqueness and regularity of the solutions to the considered problem are obtained.
The research is interesting, the system can be also interpreted as distributed and boundary control.
I read your work with great interest and pleasure. The research approaches used in the work are the Leray-Schauder degree theory, the
-theory of linear and quasi-linear parabolic equations, Green’s first identity for scalar-valued functions and continuously differentiable vector field in
dimensional space, the Lions-Peetre's interpolation methods. The fractional steps method is used to approximate the solution of the nonlinear second-order reaction-diffusion problem. The conclusion section contains future directions about the study of distributed and/or boundary nonlinear optimal control problems, numerical implementation of the conceptual algorithm, as well as various simulations regarding the physical phenomena described by nonlinear parabolic problem, considered in the manuscript. The main question of the whole work is the convergence of the sequence of solutions to the approximate problems to the unique solution of the nonlinear second-order reaction-diffusion problem.
In the present paper, the main contributions are in the obtained results concerning:
1) rigorous qualitative study is elaborated for a second-order reaction-diffusion problem;
2) the existence, a priori estimates, regularity and uniqueness of a classical solution in the Sobolev space is proved;
3) an iterative scheme of fractional steps type, associated to the nonlinear second-order reaction-diffusion problem is develop, the dependence of the solution on the physical parameters is emphasized;
4) the convergence and error estimate results for the proposed numerical scheme are established.
Although, overall manuscript is wisely written and presented in a scholarly manner, there are some suggestions which require attention of the authors.
In my point of view, this work is very well ordered and well written. The exposition is clear and correct. I believe that the submitted article should be published.
Below, the authors can find some corrections or comments that can be useful for the improvement of the paper. In particular, following should pay attention before possible acceptance of this manuscript:
- I recommend to revise the whole manuscript for grammatical mistakes.
Some technical errors: "Introducere", "boudary", " expresed", " writting ", "manupulated", etc.
- Some authors work in a specialized field, or with specialized approaches in a popular field. So it is relevant to cite themselves. I spotted that a paper is with 22 citations, around which 15 are self-citations. I understand that people do this to work on their previous studies, but from all titles in the references section in the work, between 65%-70% the authors cite himself. The question to the authors is the citations are relevant or not?
- Also, in the references section Italic is used randomly, the journal titles and book titles should be in Italic.

Author Response
Our answer to Reviewer 4:
1. The whole text of the paper was carefully revised; the typos was fixed and marked with orange color;
2. In our opinion, all citations are relevant; more over, we have added a new reference (see [23]) as suggested the reviewer 1.
3. All references was corrected according to Axioms instructions for references.
Tank you very much for reviewing our paper.
Best regards,
Constantin FETECAU and Costică MOROSANU